# Novel Somay’s GLUCAR Index Efficiently Predicts Survival Outcomes in Locally Advanced Pancreas Cancer Patients Receiving Definitive Chemoradiotherapy: A Propensity-Score-Matched Cohort Analysis

**DOI:** 10.3390/jpm14070746

**Published:** 2024-07-13

**Authors:** Erkan Topkan, Sukran Senyurek, Nulifer Kılic Durankus, Duriye Ozturk, Ugur Selek

**Affiliations:** 1Department of Radiation Oncology, Faculty of Medicine, Baskent University, Adana 01120, Turkey; 2Department of Radiation Oncology, Koc University School of Medicine, Istanbul 34450, Turkey; ssenyurek@kuh.ku.edu.tr (S.S.); ndurankus@kuh.ku.edu.tr (N.K.D.); uselek@ku.edu.tr (U.S.); 3Department of Radiation Oncology, Faculty of Medicine, Afyonkarahisar Health Sciences University, Afyonkarahisar 03030, Turkey; duriye.ozturk@afsu.edu.tr

**Keywords:** pancreatic cancer, prognosis, glucose, C-reactive protein-to-albumin ratio, GLUCAR index

## Abstract

Background: Propensity score matching (PSM) was used to investigate the prognostic value of a novel GLUCAR index [Glucose × (C-reactive protein ÷ albumin)] in unresectable locally advanced pancreatic cancer (LA-NPC) patients who received definitive concurrent chemoradiotherapy (CCRT). Methods: The PSM analysis comprised 142 LA-PAC patients subjected to definitive CCRT. Receiver operating characteristic (ROC) curve analysis was utilized to identify relevant pre-CCRT cutoffs that could effectively stratify survival results. The primary and secondary objectives were the correlations between the pre-CCRT GLUCAR measures and overall survival (OS) and progression-free survival (PFS). Results: The ROC analysis revealed significance at 43.3 for PFS [area under the curve (AUC): 85.1%; sensitivity: 76.8%; specificity: 74.2%; J-index: 0.510)] and 42.8 for OS (AUC: 81.8%; sensitivity: 74.2%; specificity: 71.7%; J-index: 0.459). Given that these cutoff points were close, the standard cutoff point, 42.8, was selected for further analysis. Comparative survival analyses showed that pre-CCRT GLUCAR ≥ 42.8 (*n* = 71) measures were associated with significantly shorter median PFS (4.7 vs. 15.8 months; *p* < 0.001) and OS (10.1 vs. 25.4 months; *p* < 0.001) durations compared to GLUCAR < 42.8 measures (*n* = 71). The multivariate analysis results confirmed the independent significance of the GLUCAR index on PFS (*p* < 0.001) and OS (*p* < 0.001) outcomes. Conclusions: Elevated pre-CCRT GLUCAR levels are robustly and independently linked to significantly poorer PFS and OS outcomes in unresectable LA-PAC patients treated with definitive CCRT.

## 1. Introduction

Pancreatic adenocarcinoma (PAC) is a deadly malignant tumor that ranks as the sixth leading cause of fatalities worldwide, with a 5-year survival rate of only 3% [1]. Surgery with negative margins is the sole prospect for a cure in PAC. Nonetheless, approximately 30% of patients are diagnosed with unresectable locally advanced PAC (LA-PAC), making them ineligible for potentially curative surgical therapies [2]. Currently, viable treatment options for LA-PAC patients include induction chemotherapy followed by reassessment for radical surgery, chemotherapy, or definitive concurrent chemoradiotherapy (CCRT). Nevertheless, the optimal course of treatment has yet to be conclusively determined, as each treatment has its benefits and challenges [3,4,5]. Regrettably, current therapies have limited effectiveness against LA-PACs, which frequently result in distant metastases (DMs) and a poor prognosis, with a median survival duration of only 9 to 13 months [6].

While the prognosis is typically discouraging, there may be a noteworthy disparity in the survival duration of LA-PAC patients, even when they receive matched treatment protocols. The long-term outcomes of the SCALOP (Selective Chemoradiation in Advanced Localized Pancreatic Cancer) trial, a multicenter, randomized, phase 2 research, are an outstanding example of this situation [7]. The median overall survival (OS) times for the capecitabine (CAP)-based and gemcitabine (GEM)-based CRT groups were 17.6 months [95% confidence interval (CI): 14.6–22.7] and 14.6 months [95% CI: 11.1–16.0], respectively. The 95% CI values show that certain patients in the CAP-CRT and GEM-CRT groups have more extended survival periods than those in the same therapy groups by 8.1 and 4.8 months, respectively. These differences in survival are significant because they correspond to 55.5% and 44.1% relative increments and imply that, even with the same treatment regimens, LA-PAC patients may significantly differ in response rates and survival times [7]. This phenomenon is consistent regardless of the patient’s initial performance statuses, local and regional disease stages, and other known prognostic factors that may appear comparable.

Significant clinical outcome contrasts among nearly identical patients might be attributed partly to the weakness of current imaging tools in detecting occult metastases [8]. Also, the current TNM (tumor–node–metastasis) staging system for non-metastatic PAC relies solely on morphological factors, such as tumor size, invasiveness to surrounding organs, and N-status, while ignoring biological markers. This disinterest in biological factors is likely the main reason for outcome disparities among otherwise comparable patients. Therefore, it is essential to identify novel features that significantly improve TNM staging and provide more precise prognostic categorization for LA-PAC patients.

Both hyperglycemia (diabetes) and chronic inflammation have been shown to impact virtually every stage of pancreatic carcinogenesis, its progression, and the response and outcomes of treatment, either independently or in combination [9,10]. Two recent meta-analyses have substantiated the correlation between diabetes and a twofold rise in the likelihood of developing PAC [11,12]. Moreover, recent research has shown that the odds of developing PAC are multiplied by 33.52 when a person simultaneously has diabetes and chronic pancreatitis, a persistent inflammatory disease [13]. Diabetic PAC is often associated with larger tumors, perineural invasion, and lower median survival durations [14,15]. Confirming these findings, Duan et al. demonstrated that high glucose levels could promote PAC progression by weakening the ability of natural killer (NK) cells to kill cancerous cells, thereby facilitating immune escape and cancer progression [16]. Similarly, previous research has also shown that measuring several inflammation indicators before therapy is associated with therapeutic outcomes in PAC patients [17,18,19,20,21,22,23,24]. One of the most potent biomarkers is the CRP-to-Alb ratio (CAR), which combines C-reactive protein (CRP) and albumin (Alb). Previous studies and meta-analyses have consistently shown that pretreatment CAR measures are associated with clinical outcomes in PAC patients, regardless of tumor stage or treatment method [25,26,27,28,29,30].

Recently, Somay et al. combined pretreatment fasting glucose and CAR measures to create a novel immune–inflammation–nutritional index, the Glucose-CAR (GLUCAR) index, for predicting tooth extraction rates after definitive CCRT in patients with locally advanced nasopharyngeal cancer [31]. In this study, the authors showed that the tooth extraction rate was significantly higher in the group with a pre-CCRT GLUCAR ≥ 31.8 (84.4% vs. 47.4% for GLUCAR ˂ 31.8; *p* < 0.001). Because high glucose levels and persistent and aggravated systemic inflammation, as indicated by a high CAR level, play critical roles in almost all PAC genesis and progression steps, we postulated that this innovative biomarker may also possess prognostic significance in LA-PAC patients after definitive CCRT. Therefore, this propensity score matching (PSM) analysis was designed to reveal the prognostic utility of Somay’s GLUCAR index in unresectable LA-PAC patients treated with definitive CCRT.

## 2. Patients and Methods

### 2.1. Patient Population

This retrospective data analysis included unresectable stage III (T4N0-1M0 per the Union for International Cancer Control/American Joint Committee on Cancer staging system 8th edition) LA-PAC patients treated with definitive concurrent CRT at the Baskent University Medical Faculty Department of Radiation Oncology from January 2010 to January 2022. The diagnosis of PAC was established through a comprehensive histopathologic examination of the tissue samples. The diagnostic and staging assessments were executed in compliance with established procedures, as previously documented [32]. To be eligible for the study, patients had to meet the following additional requirements: age 18 to 80 years, Karnofsky performance score (KPS) 70–100, no previous chemotherapy/radiotherapy history, adequate bone marrow (hemoglobin value of ≥10 g/dL, leucocyte of ≥4.000 μL, and thrombocyte of ≥100.000 μL) and hepatic (aspartate aminotransferase or alanine aminotransferase of <5 times the upper limit) and renal (serum creatinine < 2 mg/dL) functions, body mass index (BMI) > 20 kg/m^2^, available records of radiotherapy and chemotherapy, and available complete blood count and biochemistry test results obtained at the first day of CRT. Patients with a history of chronic immunosuppressive medication or steroid usage, chronic inflammatory diseases, active chronic or acute infections, radiation hypersensitivity syndromes, or blood transfusions within 90 days before CRT initiation were excluded from this study (Figure 1). 

### 2.2. Treatment Protocol

Each patient in this study received definitive CCRT with one to two courses of cisplatin (*n* = 38), oral capecitabine (*n* = 35), continuously infused 5-fluorouracil (*n* = 29), gemcitabine (*n* = 21), or cisplatin-based doublet chemotherapy (*n* = 19) administered concurrently with radiotherapy. The target volumes were defined and delineated as described in our previous report [33]. In brief, the gross tumor volume for each patient comprised the primary tumor, as well as lymph nodes that were visible on contrast-enhanced computerized tomography (CT) scans (with a short axis > 1.0 cm) and/or fluorodeoxyglucose positron-emission tomography (FDG-PET) images. Nodes that measured <1.0 cm were determined to be tumor-positive only if they were found to be metabolically active (with a maximum standard uptake value of >2.5) on the FDG-PET scan. A uniform total dose of 45 Gy encompassing the defined planning target volume was delivered over five weeks (1.8 Gy/fraction, five days per week). This study did not permit elective nodal irradiation, as it did not comply with institutional standards for LA-PAC patients. After completing CCRT, all patients were recommended to have an additional 4–6 cycles of maintenance gemcitabine (*n* = 67) or 2–4 cycles of cisplatin-based doublet (*n* = 75) chemotherapy. Supportive care measures included antiemetic medication, hydration, and nutritional supplements, as necessary.

### 2.3. GLUCAR Index Calculation and Measurement

The GLUCAR index was calculated using the original formula created by Somay et al. [31]: GLUCAR = [Fasting glucose (mg/dL) × CRP (mg/dL) ÷ Alb (g/dL)], where glucose, CRP, and Alb represent the pretreatment data obtained from the standard blood biochemistry tests carried out on the first day of concurrent CRT. Each parameter of the GLUCAR index was measured using the Abbott Architect c8000 Biochemistry Autoanalyzer following the manufacturer’s instructions (Abbott Architect c8000 Biochemistry Autoanalyzer, Abbott, Chicago, IL, USA) [33].

### 2.4. Treatment Response Evaluation

Patients had examinations at a frequency of every 3 months during the first 2 years, with intervals of 6 months between the 3rd and 5th years, and annually after that, or more often if necessary. The assessment of treatment response was first conducted 3 months after CCRT using restaging FDG-PET-CT and magnetic resonance imaging (MRI)/CT scans, following the criteria established by the European Organization for Research and Treatment of Cancer (EORTC) in 1999. Every patient had follow-up via total blood count and biochemistry tests, serum CA 19-9 concentrations, and FDG-PET-CT scans until a complete metabolic response was verified. In instances where complete metabolic response was confirmed, abdomen MRI/CT scans substituted the FDG-PET-CT imaging. Patients were only subjected to additional evaluations if deemed necessary, such as abdominal ultrasonography, chest CT, cranial MRI, bone scintigraphy, endoscopic examinations or open exploration.

### 2.5. Statistical Analysis

The primary aim of this PSM analysis was to assess the probable correlation between Somay’s GLUCAR index’s pre-CCRT measures and OS. In this study, OS refers to the interval between the first day of concurrent CCRT and the death or last follow-up dates. The secondary objective was to evaluate progression-free survival (PFS): the interval between the first day of CCRT and the date of any disease progression, last visit, or death, whichever comes first. Medians and ranges were used to assess continuous data, whereas frequency distributions were utilized to describe categorical variables. Frequency distributions were compared using chi-square, Student’s *t*-test, Pearson’s exact test, or Spearman’s correlation estimates as necessary. The present study employed receiver operating characteristic (ROC) curve analysis to determine the feasibility of a pre-CCRT GLUCAR cutoff that could potentially stratify the study population into two subgroups with significantly distinct OS and PFS outcomes. We analyzed the potential impact of various risk factors on the results of OS and PFS using Kaplan–Meier estimates and log-rank tests. Only those factors that showed significance in the initial univariate comparisons were included in the multivariate Cox proportional hazard model to assess the potential interactions between these variables and survival outcomes. All comparisons were two-sided, and *p*-values < 0.05 were deemed statistically significant. Bonferroni correction and associated *p*-values were used to limit the accidental false-positive results for simultaneously comparing the outcomes between three or more groups. As our investigation was a retrospective analysis, we used propensity scores to ensure comparability across the GLUCAR groups and minimize biases in the findings. To accomplish this goal, we considered several factors, including age, gender, Karnofsky performance score, tumor histology, N-stage, CA 19-9 measurements, and concurrent and maintenance treatment cycles. We established 1:1 matched groups using nearest-neighbor matching with logistic regression. The caliper was set to 0.2, and replacement was not allowed.

## 3. Results

Throughout the study period, our department evaluated 289 individuals diagnosed with LA-PAC. However, 32, 24, 9, 6, and 5 of them were excluded from the study for the following reasons: receiving induction chemotherapy, having DM during the staging procedure, having lower performance scores (KPS < 70), being unwilling to undergo concurrent chemotherapy, and being unable to complete the planned CCRT course. Therefore, the present study consisted of a cohort of 217 participants who had CCRT as part of their treatment. Table 1 outlines the baseline patient and disease characteristics. The patients had a median age of 57, ranging from 39 to 77. The majority, 76.5%, were male. The most frequent tumor location was the pancreatic head (80.1%), while 47.5% had a lymph node status of N1-2. Before commencing the treatment, 96 patients (44.4%) had a confirmed diagnosis of diabetes mellitus, 112 (51.6%) had severe weight loss (>5% in the last 6 months), and 121 (55.8%) had CA 19-9 levels > 90 IU/Ml according to a previously established cutoff in the benchmark Charité Onkologie 001 (CONKO-001) randomized trial [34]. 

Upon completing the final analysis, the median duration of follow-up was 20.3 months, with a range of 3.4 to 137.6 months. Of the 217 patients analyzed, 51 (23.5%) were still alive, while 26 (12.0%) showed no disease progression. The median and 5-year PFS estimates were 7.5 months (CI: 5.6–9.4 months) and 10.0%, respectively. The corresponding estimates for median and 5-year OS rates were 17.4 months (95% CI: 14.7–20.2) and 12.9%, respectively. Among 166 deaths, 153 (92.2%) were attributed to uncontrolled disease progression, with 142 (85.5%) due to widespread DM and 11 (6.7%) due to isolated locoregionally progressive disease. 

The search for a possible GLUCAR cutoff that may interact with treatment outcomes via ROC curve analysis revealed significance at 43.3 for PFS [area under the curve (AUC): 85.1%; sensitivity: 76.8%; specificity: 74.2%; J-index: 0.510] and 42.8 for OS (AUC: 81.8%; sensitivity: 74.2%; specificity: 71.7%; J-index: 0.459), as depicted in Figure 2. Given the proximity of the two cutoffs, the standard cutoff of 42.8 was selected for further analysis. Hence, based on their GLUCAR measures, the patients were categorized into two groups: Group 1: GLUCAR < 42.8 (*n* = 86) and Group 2: GLUCAR ≥ 42.8 (*n* = 131). 

A PSM 1:1 analysis was conducted in the succeeding stage to determine the matched GLUCAR groups based on all variables listed in Table 1. As a result, out of 217 patients, 142 were matched, with each GLUCAR group comprising 71 patients (<42.8 vs. ≥42.8). The subsequent data and results will exclusively represent those derived from the whole PSM cohorts. The two PSM GLUCAR groups had similar distributions of baseline characteristics, as shown in Table 1. The data presented below display the outcomes of PSM GLUCAR cohorts. The entire PSM cohort’s median follow-up duration was 20.7 months (range: 3.8–137.6 months). The Kaplan–Meier survival estimates showed that a GLUCAR ≥ 42.8 measure before treatment was linked to significantly shorter median PFS (4.7 vs. 15.8 months; *p* < 0.001) and OS (10.1 vs. 25.4 months; *p* < 0.001) durations compared to a GLUCAR < 42.8 value (Figure 2). The corresponding 3-year (19.4% vs. 10.2%) and 5-year (19.4% vs. 5.1%) PFS, and 3-year (36.8% vs. 10.8%) and 5-year (27.3% vs. 5.4%) OS rates were also numerically inferior in the GLUCAR ≥ 42.8 cohort, indicating that the long-term outcomes were durably worsening with a high GLUCAR measure (Figure 3). Given the median PFS of the GLUCAR ≥ 42.8 cohort being only 4.7 months, we also examined the cause(s) of this finding. Upon further analysis of this patient group, it became clear that out of 71 patients, 57 (80.3%) experienced early DM, with 16 (22.6%) developing DM within 3 months of follow-up and 41 (57.7%) between 3 and 6 months. 

The results of the univariate analyses revealed that, apart from the GLUCAR ≥ 42.8 group (vs. <42.8), patients presenting with a KPS of 70–80 (vs. 90–100), WL > 5% (vs. ≤5%), CA19-9 ≥ 90 U/m/L (vs. <90 U/m/L), and an N-stage of 2 (vs. 0–1) were the predictors associated with poor PFS (*p* < 0.05 for each) and OS (*p* < 0.05 for each) outcomes, respectively (Table 2 and Table 3). The multivariate analyses confined to these factors confirmed the individual significance of each factor on the PFS (*p* < 0.05 for each) and OS (*p* < 0.05 for each) outcomes (Table 2)

## 4. Discussion

LA-PAC patients often have a poor prognosis due to their relative resistance to conventional cancer therapies such as radiotherapy and chemotherapy. Nevertheless, significant disparities in outcomes among LA-PAC patients can still occur despite receiving indistinguishable treatments. Since PAC initiation and progression steps are established to be strongly associated with hyperglycemia and chronic inflammation, we hypothesized that the recently introduced Somay’s GLUCAR index might help predict disease outcomes in unresectable LA-PAC patients undergoing definitive CCRT. Within this context, results of our current PSM analysis demonstrated that pretreatment GLUCAR ≥ 42.8 was an independent and robust predictor of poor PFS (4.7 vs. 15.8 months for GLUCAR < 42.8; *p* < 0.001) and OS (10.1 vs. 25.4 months for GLUCAR < 42.8; *p* < 0.001) outcomes in LA-PAC patients treated with definitive CCRT. Moreover, inferior 3-year and 5-year PFS and OS rates in the GLUCAR ≥ 42.8 group suggested that the novel GLUCAR index has long-term discriminatory power and could potentially serve as a robust indicator of disease trajectory.

The present investigation has documented further evidence regarding the poor prognostic significance of well-established N1-2-stage, CA 19-9 levels ≥ 90 U/m/L, and WL > 5% at presentation [34,35,36]. However, the most influential finding of our study was the exhibit of pretreatment GLUCAR levels ≥ 42.8 as a novel, independent, and robust indicator of poorer PFS (*p* < 0.001) and OS (*p* < 0.001) in unresectable LAPAC patients who underwent exclusive CCRT. Hyperglycemia-aggravated systemic inflammation, suppressed immunity, and poor nutritional status play crucial roles in every step of the genesis and progression of PACs [37]. Considering these facts together with its achievability without excess cost, simple calculation formula, reproducibility, and long-term durability, the present results suggest that the novel GLUCAR index, which integrates pretreatment glucose and CAR levels, may be a brand-new biologic marker in the prognostic stratification of LAPAC patients. The exact causalities behind the strong correlation between high GLUCAR levels before CCRT and significantly lower PFS and OS rates remain unclear. However, even though comparable research is unavailable, we can analyze GLUCAR’s separate ingredients, namely glucose and CAR, to derive plausible explanations for this correlation.

It has been established that high glucose levels (hyperglycemia), the first component of the novel GLUCAR index, may either cause PAC or be induced by PAC [12,38,39]. Hyperglycemia can contribute to tumor-related inflammation by triggering the release of pro-inflammatory cytokines such as interleukin-6 (IL-6), interferon γ (IFN-γ), tumor necrosis factor-alpha (TNF-α), and resistin [40]. This intricate cascade of events can ultimately result in mitochondrial dysfunction, oxidative stress, the buildup of lipids within liver or skeletal muscle cells, and a decrease in β-oxidation. These events may lead to the development of insulin resistance and the activation of downstream carcinogenic signaling pathways, including NF-κB, JNK/MAPK, and c-Jun [41,42]. Hyperglycemia may also cause immune dysregulation by diminishing the functions of immune cells infiltrating cancer tissues, including CD8+ T cells, neutrophils, and myeloid-derived suppressor cells (MDSCs); reprogram MDSCs to regulate M1 and M2 differentiation; and stimulate IL-6 secretion by inducing TNF-α secretion by monocytes and macrophages [43,44]. These hyperglycemia functions may lead to aggravated inflammation, immune suppression, tumor growth, tumor progression, and infiltration [44,45]. It has been observed that hyperglycemia can play a protective role in impeding apoptosis in tumor cells, which is brought about by the specific inhibition of Serine 46 phosphorylation, a known activator of p53, and cytochrome C-mediated apoptosis by glucose-induced increased glutathione (GSH) synthesis in tumor cells [46,47]. Hyperglycemia may induce or accelerate malignant behavior by increasing reactive oxygen species production in a concentration-dependent manner, with accompanying increases in urokinase fibrinogen activator and superoxide dismutase-dependent hydrogen peroxide [48]. High glucose levels may additionally promote cancer stem cell properties via activation of the transforming growth factor β (TGF-β) signaling pathway, thus synergistically increasing tissue fibrosis, cell invasion, migration, and metastasis in tumor cells, creating a treatment-resistant PDAC phenotype [49]. Excess glucose may bind non-enzymatically to amino groups on nucleic acids, lipids, and proteins, forming precursors of advanced glycation end products (AGEs), the degree of which is related to the severity and duration of hyperglycemia [50]. Hyperglycemia causes long-term AGE buildup and increases intracellular inflammatory signals, which promotes NF-κB activation and oxidative stress, ultimately leading to carcinogenesis initiation and advancement [51]. Based on the mechanisms we have discussed, our current research findings suggest that PDAC patients with elevated blood glucose levels may be susceptible to a treatment-resistant strain, leading to a discouraging prognosis. Our results are consistent with prior studies indicating that unfavorable outcomes for PDAC can be observed across different treatment options in hyperglycemic patients [52,53,54,55].

The CAR component of the GLUCAR index consists of CRP and albumin. CRP is a protein manufactured in the liver and activated by pro-inflammatory cytokines during the acute phase of an immune response. CRP is widely acknowledged as a nonspecific but robust and dependable marker of systemic inflammation in the host. The upregulation of CRP production is invariably accompanied by a rapid decrease in blood albumin levels, leading to hypoalbuminemia [56]. This inverse relationship between CRP and albumin levels is attributed mainly to the suppression of albumin synthesis in the hepatocytes caused by CRP and its byproducts, tumor necrosis factor-alpha (TNF-α) and interleukin-6 (IL-6), which can be utilized in monitoring the severity of inflammatory reactions in various pathological conditions, including solid cancers. Elevated CRP and reduced albumin levels also indicate a pre-cachectic/cachectic state in patients [57,58]. This state is strongly associated with an overtly stimulated systemic inflammatory condition and poor prognosis in many solid cancers, including the LAPACs [35]. Following our present investigation, antecedent research studies and meta-analysis findings have provided empirical evidence that substantiates the significant predictive ability of the CAR component of the GLUCAR index. These studies have demonstrated that a high level of CAR before treatment was correlated with an unfavorable prognosis for PDAC, independent of disease stage or treatment modality, which was superior to other peripheral blood cell count-based biomarkers [30,59,60,61].

Our analysis also revealed a noteworthy finding: GLUCAR > 42.8 resulted in PFS (median 4.7 months) and OS (median 10.1 months) results that closely resembled those of patients with metastatic PAC who received palliative systemic chemotherapy. An analysis of the likely causes of this negative result revealed that 57 out of 71 patients (80.3%) developed DM. Of these patients, 16/57 (28.1%) experienced DM only within 3 months of follow-up, while 41/57 (71.9%) experienced DM within 6 months, and the reason for death was metastatic disease progression in all patients (100%). These results strongly and reasonably indicate occult DMs before the start of CCRT, which were not detectable by the existing staging methods, such as MRI and PET-CT, due to their limited resolution. Additionally, they were not responsive to the chemotherapeutic treatment used in our research. The present indication is consistent with research findings that suggest a correlation between inflammation and an increased risk of DM due to the development of chemotherapy and radiotherapy resistance in index cancer. This resistance occurs through several associated mechanisms, including uninhibited autophagy during aggravated inflammatory conditions, facilitated immune evasion, persistent and exacerbated inflammation and immune dysregulation caused by hyperglycemia, hyperglycemia-induced protection of tumor cells against apoptosis, poor nutritional status indicated by a high CAR value, and the activation of multiple metabolic pathways that promote the rapid growth and proliferation of tumor cells, which may promote tumor invasion and metastasis [47,62,63]. Irrespective of the root cause, our research findings emphasize the urgent need for implementing more advanced staging tools and more potent systemic therapies to the staging and treatment algorithms of these patients. In this context, liquid biopsies obtained from plasma or wash samples via endoscopic ultrasound-guided fine-needle biopsy may help detect high-risk LAPAC patients for early DM [64]. Besides this maneuver, it may be wise to initiate induction chemotherapy first and withhold aggressive CCRT for those who remain free of DM after systemic treatment. Upon further validation through research, GLUCAR may ascertain the requisite intensity of therapies, potentially enhancing patient prognoses by tailoring treatments in some patients while averting futile interventions in others deemed resistant to presently available treatment modalities.

The present study employed the widely recognized PSM analysis methodology to counterbalance the inhomogeneities between the two GLUCAR groups; however, it is still crucial to acknowledge certain limitations. First, it is a retrospective study, analyzing a cohort from a single institution. As such, unanticipated biases may have inadvertently influenced the results. Second, while the two GLUCAR cohorts shared similar demographic and treatment characteristics, discrepancies in the adjuvant and rescue treatments may have provided an unpredictable advantage to one group regarding tumor control and survival outcomes. Third, due to the planned inclusion of only T4N0-2M0 stage patients with a good performance status (KPS 70–100) and the sole use of definitive CCRT in all patients, the present findings may only reflect part of the real-world practices in LA-PAC patients. Therefore, it is necessary to address the discriminatory potential of the GLUCAR index in patients who are at an early or metastatic stage, patients who have a lower KPS, and patients who are managed with other treatment choices, such as chemotherapy with or without surgery, in these specific scenarios. Fourth, this study exclusively examined the pre-CCRT glucose, CRP, and albumin measures before treatment initiation. Notably, these biochemical markers are susceptible to substantial fluctuations throughout the CCRT and follow-up phases due to variations in tumor load, systemic inflammatory status, and host immunity. Therefore, further research into the kinetics of these variables may offer valuable insights into identifying potentially more reliable cutoffs for each of them and, consequently, the resulting GLUCAR index. Such findings could help refine the accuracy of this index and enhance its clinical utility. Fifth, we did not investigate the possible association between the FDG uptake values and baseline GLUCAR levels, which might provide valuable perspicuity into the tumor’s biological characteristics. Finally, despite using PSM analysis to reduce the diversity between the two GLUCAR cohorts, the findings presented here should be cautiously interpreted and valued as hypothetical rather than solid guidance for all LA-PAC patients treated with CCRT until the results of well-designed studies in larger cohorts substantiate them.

## 5. Conclusions

In this study, higher GLUCAR levels before treatment were independently linked to significantly poorer PFS and OS outcomes in unresectable LA-PAC patients who underwent definitive CCRT. These findings imply that GLUCAR could become a valuable prognostic tool for identifying high-risk patients and helping select the most appropriate treatment, provided that future research confirms these findings.

## Figures and Tables

**Figure 1 jpm-14-00746-f001:**
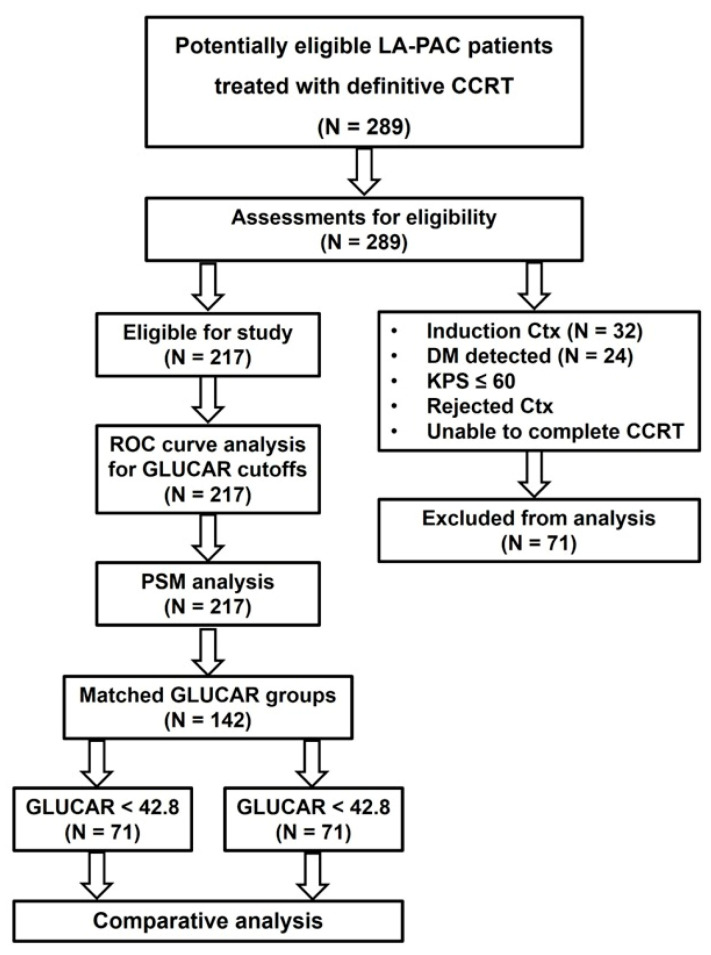
Flowchart showing the patient eligibility and analytical characteristics of this study. Abbreviations: LA-PA: locally advanced pancreatic adenocarcinoma; CCRT: concurrent chemoradiotherapy; Ctx: chemotherapy; DM: distant metastasis; KPS: Karnofsky performance score; PSM: propensity score matching; GLUCAR: [Glucose × (C-reactive protein ÷ albumin)].

**Figure 2 jpm-14-00746-f002:**
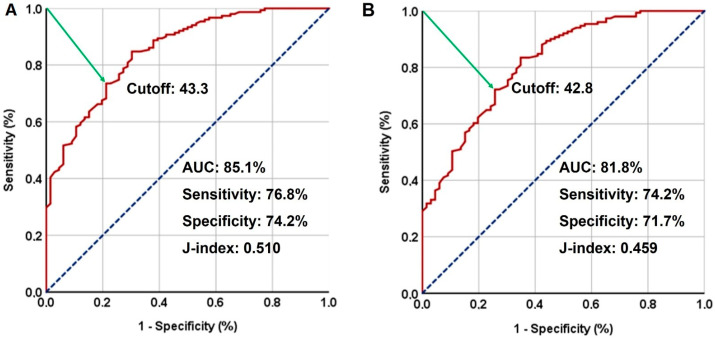
Outcomes of receiver operating characteristic (ROC) curve analyses and survival outcomes per GLUCAR index group: (**A**) progression-free survival: cutoff: 43.3; area under the curve (AUC): 85.1%; sensitivity: 76.8%; specificity: 74.2%; J-index: 0.510. (**B**) Overall survival cutoff: 42.8; AUC: 81.8%; sensitivity: 74.2%; specificity: 71.7%; J-index: 0.459.

**Figure 3 jpm-14-00746-f003:**
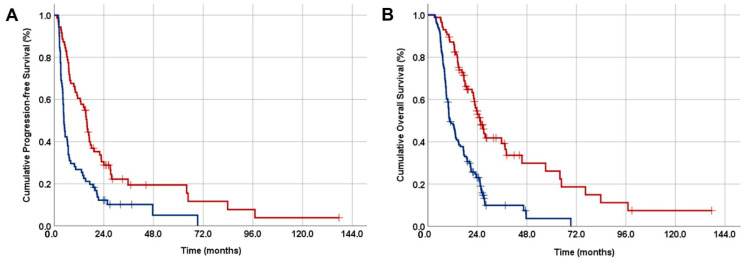
Results of comparative Kaplan–Meier survival estimates per GLUCAR index group (red lines: GLUCAR < 42.8, and blue lines: GLUCAR ≥ 42.8): (**A**) progression-free survival, and (**B**) overall survival. Abbreviation: GLUCAR = [Glucose × (C-reactive protein ÷ albumin)].

**Table 1 jpm-14-00746-t001:** Baseline patient and disease characteristics for the entire study group and per systemic immune–inflammation index subgroups.

Characteristic	All Patients	GLUCAR < 42.8	GLUCAR ≥ 42.8	*p*-Value	PSM Patients	GLUCAR < 42.8	GLUCAR ≥ 42.8	*p*-Value
	(*n* = 217)	(*n* = 86)	(*n* = 131)		(*n* = 142)	(*n* = 71)	(*n* = 71)	
Median age, years (range)	57 (39–77)	56 (39–68)	57 (39–77)	0.97	56 (39–77)	56 (39–77)	56 (39–77)	1.0
Age group, *n* (%)				0.87				0.91
<70 years	171 (78.8)	70 (81.4)	101 (80.6)		113 (79.6)	57 (80.3)	56 (78.9)	
≥70 years	46 (21.2)	16 (18.6)	30 (19.4)		29 (20.4)	14 (19.7)	15 (21.1)	
Gender, *n* (%)				0.41				0.68
Female	51 (23.5)	22 (25.6)	29 (22.1)		24 (16.9)	13 (18.3)	11 (15.5)	
Male	166 (76.5)	64 (74.4)	102 (77.9)		118 (83.1)	58 (81.7)	60 (84.5)	
KPS, *n* (%)				0.53				1.0
90–100	176 (81.1)	67 (77.9)	109 (83.2)		118 (83.1)	59 (83.1)	59 (83.1)	
70–80	41 (18.9)	19 (22.1)	22 (16.8)		24 (16.9)	12 (16.9)	12 (16.9)	
WL > 5%				0.02				0.83
Absent	112 (51.6)	38 (44.2)	74 (56.4)		68 (47.9)	35 (49.3)	33 (46.5)	
Present	105 (48.4)	48 (55.8)	57 (43.6)		74 (52.1)	36 (50.7)	38 (53.5)	
Tumor location, *n* (%)				0.78				0.87
Head	176 (81.1)	68 (79.1)	108 (82.4)		116 (81.7)	59 (83.1)	57 (80.3)	
Body/tail	41 (18.9)	18 (20.9)	23 (17.6)		26 (18.3)	12 (16.9)	14 (19.7)	
N stage, *n* (%)				0.38				0.79
0	114 (52.5)	47 (54.6)	67 (51.1)		74 (52.1)	36 (50.7)	38 (53.5)	
1–2	103 (47.5)	39 (45.4)	64 (48.9)		68 (47.9)	35 (49.3)	33 (46.5)	
CA 19-9 status, *n* (%)				0.17				0.81
≤90 U/mL	96 (44.2)	42 (48.8)	54 (41.2)		66 (46.5)	34 (47.9)	32 (45.1)	
>90 U/mL	121 (55.8)	44 (51.2)	77 (58.8)		76 (53.5)	37 (52.1)	39 (54.9)	

Abbreviations: GLUCAR: [Glucose × (C-reactive protein ÷ albumin)]; PSM: propensity score matched; KPS: Karnofsky performance score; WL: weight loss; N-stage: nodal stage; CA 19-9: cancer antigen 19-9. Note: A Chi-Square test was employed to perform cross-tabulated comparisons for all parameters.

**Table 2 jpm-14-00746-t002:** Outcomes of uni- and multivariate analysis for the propensity-score-matched cohort.

Factor	Overall Survival			Progression-Free Survival		
	Univariate *p*-Value	Multivariate *p*-Value	HR (95% CI)	Univariate *p*-Value	Multivariate *p*-Value	HR
Age group (<70 vs. ≥70 years)	0.82	-	0.96 (0.88–1.06)	0.78	-	00.96 (0.82–1.18)
Gender (female vs. male)	0.53	-	0.94 (0.83–1.11)	0.67	-	0.89 (0.72–1.17)
KPS (90–100 vs. 70–80)	0.009	0.014	0.81 (0.66–0.95)	0.007	0.011	0.72 (0.56–0.88)
WL >5% (No vs. Yes)	<0.001	<0.001	0.67 (0.53–0.81)	<0.001	<0.001	0.62 (0.49–0.74)
Tumor location (H vs. B/T)	0.73	-	0.93 (0.86–1.07)	0.69	-	0.84 (0.64–1.09)
N-stage (0–1 vs. 2)	0.004	0.007	0.76 (0.61–0.87)	0.004	0.005	058 (0.39–0.78)
CA19-9 (<vs. ≥90 U/m/L)	<0.001	<0.001	0.71 (0.51–0.89)	<0.001	<0.001	0.63 (0.49–0.75)
GLUCAR (<vs. ≥42.8)	<0.001	<0.001	0.24 (0.13–0.37)	<0.001	<0.001	0.32 (0.021–0.44)

Abbreviations: HR: hazard ratio; CI: confidence interval; KPS: Karnofsky performance score; WL: weight loss; H: head; B/T: body/tail; N-stage: nodal stage; CA 19-9: cancer antigen 19-9; GLUCAR: [Glucose × (C-reactive protein ÷ albumin)]. Note: Kaplan–Meier estimates and log-rank tests were used for intergroup comparisons. Cox regression analysis was used for multivariate comparisons.

**Table 3 jpm-14-00746-t003:** Survival results according to the factors exhibiting independent prognostic significance in multivariate analyses.

Endpoint	All Patients	KPS			N-Stage			CA 19-9 Status			GLUCAR Index			>5% WL		
		90–100	70–80	*p*-Value	0–1	2	*p*-Value	<90 U/m/L	≥90 U/m/L	*p*-Value	<42.8	≥42.8	*p*-Value	Absent	Present	*p*-Value
	*n* = 142	(*n* = 118)	(*n* = 24)		(*n* = 74)	(*n* = 68)		(*n* = 66)	(*n* = 76)		(*n* = 71)	(*n* = 71)		(*n* = 68)	(*n* = 74)	
PFS				0.011			0.005			<0.001			<0.001			<0.001
Median (mos.)	7.5	13.7	5.9		9.1	6.4		11.3	5.7		15.8	4.7		12.1	5.2	
3-year (%)	16.8	20.5	6.8		27.1	9.8		24.5	8.3		19.4	10.2		23.4	7.9	
5-year (%)	12.9	14.1	0		21.2	4.9		17.1	4.8		19.4	5.1		18.2	4.9	
OS				0.014			0.007			<0.001			<0.001			<0.001
Median (mos.)	17.4	19.0	4.6		23.1	8.9		20.8	8.7		25.4	10.1		22.2	15.1	
3-year (%)	24.7	41.7	5.6		37.2	10.9		27.4	16.9		36.8	10.8		33.7	16.8	
5-year (%)	10.0	14.1	0		18.9	4.2		15.7	5.4		27.3	5.4		24.2	7.3	

Abbreviations: KPS: Karnofsky performance score; N-stage: nodal stage; CA 19-9: cancer antigen 19-9; GLUCAR: [Glucose × (C-reactive protein ÷ albumin)]; WL: weight loss; PFS: progression-free survival; OS: overall survival. Note: Kaplan–Meier estimates and log-rank tests were used for intergroup comparisons.

## Data Availability

The presented data belong to and are stored at the Baskent University Faculty of Medicine; they cannot be shared without permission. For researchers who meet the criteria for access to confidential data, contact the Baskent University Corporate Data Access/Ethics Board: adanabaskent@baskent.edu.tr.

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
