# Peer review of "Novel Somay’s GLUCAR Index Efficiently Predicts Survival Outcomes in Locally Advanced Pancreas Cancer Patients Receiving Definitive Chemoradiotherapy: A Propensity-Score-Matched Cohort Analysis"

_jpm, 2024, doi:10.3390/jpm14070746_

Round 1

Reviewer 1 Report

Comments and Suggestions for Authors

The manuscript entitled “Novel Somay’s GLUCAR Index Efficiently Predicts Survival Outcomes in Locally Advanced Pancreas Cancer Patients Receiving Definitive Chemoradiotherapy: A Propensity-Score-Matched Cohort Analysis” provides valuable data on the role of the GLUCAR score in stratifying pancreatic cancer patients based on their risk for treatment failure. The study should be accepted but with the following corrections.

In lines 55 and 59, please add reference number 7 at the end of the corresponding sentences.

In line 111, please include criteria for adequate bone marrow, hepatic, and renal functions.

Lines 185-188: A flowchart might be a better way to present these data.

In the second sentence of the Results section, it is mentioned that 24 patients were excluded due to having DM during the staging procedure. Later in the same paragraph, it is stated that 96 patients (44.4%) had a confirmed diagnosis of DM. Please clarify who these 96 patients are.

The table captions should be placed above the tables (e.g., Table 1). In Table 1, the name of the statistical test used should be mentioned either as part of the table or in the abbreviations.

Figure captions should be placed below the figures. The cut-off point should be marked on the figure 1 and indicated in the caption.

Lines 243-245: Upon further analysis of this patient group, it became evident that 71 patients experienced early DM. Of these, 16 (22.5%) developed DM within 3 months of follow-up, 41 (57.7%) within 6 months, and the remaining 14 later on. This sentence is unclear; 71 in total experienced DM, 16 within 3 months, 41 within 6 months, and the remaining 14 later on?

Figure 2 caption should be below the figure.

When comparing patients with a GLUCAR score below and above 42.8, statistical significance was found only for WL. Why was it necessary to match patients according to all parameters as part of the PMS analysis? Also, was the cut-off value obtained from the whole group (217 patients) and then applied to the PMS group (142 patients)?

The names of the statistical tests are missing in all tables.

The caption for Table 2 should be placed above the table.

It is unclear whether the role of ECOG PS score of 2 (versus 0-1), WL > 5% (versus ≤ 5%), CA19-9 ≥ 90 U/mL (versus < 90 U/mL), and an N-stage of 2 were obtained from the PMS group or the whole group according to OS and PFS. Although Table 3 clarifies that it is based on the group of 142 patients, this should be mentioned earlier.

The entire Table 3 is not visible.

Line 394 states that patients had a good performance status (ECOG 0-1), but the Results section includes patients with PS 2 as well. Please clarify.

Author Response

Reviewer 1

We thank Reviewer 1 for providing invaluable comments that will undoubtedly strengthen the comprehensiveness of our manuscript.

Comment 1. In lines 55 and 59, please add reference number 7 at the end of the corresponding sentences.

Response 1. We included reference number 7 in the specified sentences of the revised manuscript, as instructed. 

Comment 2. In line 111, please include criteria for.

Response 2. As per recommendations, we have included the necessary criteria for adequate bone marrow, hepatic, and renal functions in the revised manuscript’s Methods section.

Comment 3. Lines 185-188: A flowchart might be a better way to present these data.

Response 3. We created a flowchart and integrated it into the manuscript as recommended.

 Comment 4. In the second sentence of the Results section, it is mentioned that 24 patients were excluded due to having DM during the staging procedure. Later in the same paragraph, it is stated that 96 patients (44.4%) had a confirmed diagnosis of DM. Please clarify who these 96 patients are.

Response 4.  We extend our apologies for any potential confusion. In our manuscript, the abbreviation DM pertains to distant metastasis. We want to clarify that in the revised manuscript, we have replaced the specified reference of DM with diabetes mellitus. Thank you for your understanding and attention to this matter.

Comment 5. The table captions should be placed above the tables (e.g., Table 1). In Table 1, the name of the statistical test used should be mentioned either as part of the table or in the abbreviations.

Response 5. The recommended changes have been made in accordance with the journal's requirements. The "Note" heading now includes the statistical tests that were used. 

Comment 6. Figure captions should be placed below the figures. The cut-off point should be marked on the figure 1 and indicated in the caption.

Response 6. The recommended changes have been made in accordance with the journal's instructions. The cutoff point is marked in revised Figure 1.

Comment 7. Lines 243-245: Upon further analysis of this patient group, it became evident that 71 patients experienced early DM. Of these, 16 (22.5%) developed DM within 3 months of follow-up, 41 (57.7%) within 6 months, and the remaining 14 later on. This sentence is unclear; 71 in total experienced DM, 16 within 3 months, 41 within 6 months, and the remaining 14 later on?

Response 7.  This phrase has been clarified as follows: Upon further analysis of this patient group, it became clear that out of 71 patients, 57 (80.3%) experienced early DM, with 16 (22.6%) developing DM within 3 months of follow-up and 41 (57.7%) between the 3 to 6 months.

Comment 8. Figure 2 caption should be below the figure.

Response 8. The recommended changes have been made in accordance with the journal's instructions.

Comment 9. When comparing patients with a GLUCAR score below and above 42.8, statistical significance was found only for WL. Why was it necessary to match patients according to all parameters as part of the PMS analysis? Also, was the cut-off value obtained from the whole group (217 patients) and then applied to the PMS group (142 patients)?

Response 9. Propensity score matching (PSM) is a quasi-experimental methodology that uses statistical techniques to establish an artificial control group by matching each treated unit with a non-treated unit with similar characteristics. By leveraging these matches, researchers can gauge the impact of an intervention. Matching proves to be a valuable tool in data analysis for evaluating the effects of an event, particularly when ethical or logistical constraints preclude randomization. The efficacy of a propensity score matching (PSM) analysis is contingent upon the integration of specific characteristics into the analysis, excluding the tested parameter or its constituents (glucose, CRP, albumin, and CAR in our investigation). Specifically, including as many variables as possible results in enhanced efficacy of the PSM analysis and more remarkable similarity among PSM groups. Therefore, we included as many parameters as possible in the PSM analysis to avoid unpredictable biases among the resultant PSM groups related to the unaccounted parameters' impact on the outcomes

In compliance with the methodology outlined in the Methods and Results sections, PSM analysis was performed on all eligible patients, excluding GLUCAR and its components. This exclusion was necessitated by the requirement for a categorical parameter in generating PSM groups. Hence, the GLUCAR cutoff utilized in our study was determined through ROC curve analysis of all eligible patients included in the PSM analysis (N=211).

Comment 10. The names of the statistical tests are missing in all tables.

Response 10. The "Note" heading now includes the statistical tests that were used.

 Comment 11. The caption for Table 2 should be placed above the table.

Response 11. The recommended changes have been made in accordance with the journal's instructions.

 Comment 12. It is unclear whether the role of ECOG PS score of 2 (versus 0-1), WL > 5% (versus ≤ 5%), CA19-9 ≥ 90 U/mL (versus < 90 U/mL), and an N-stage of 2 were obtained from the PMS group or the whole group according to OS and PFS. Although Table 3 clarifies that it is based on the group of 142 patients, this should be mentioned earlier.

Response 12. ‘The subsequent data and results will exclusively represent those derived from the whole PSM cohorts’. This phrase has been added to the Results section of the revised manuscript for clarity.

Comment 13. The entire Table 3 is not visible.

Response 13. Due to Table 3's considerable size, this outcome was unavoidable. Nevertheless, should the manuscript be accepted, we trust that the journal's proficient editorial team will appropriately arrange it.

Comment 14. Line 394 states that patients had a good performance status (ECOG 0-1), but the Results section includes patients with PS 2 as well. Please clarify.

Response 14.  Thanks a lot for the kind reminder. As delineated in the Methods section, we employed KPS instead of ECOG. Subsequently, this mistake has been fixed throughout the entire manuscript.

Reviewer 2 Report

Comments and Suggestions for Authors

The manuscript "Novel Somay's GLUCAR Index Efficiently Predicts Survival Outcomes in Locally Advanced Pancreas Cancer Patients Receiving Definitive Chemoradiotherapy: A Propensity-Score-Matched Cohort Analysis" by Topkan, et al. presents a unique biomarker to assess locally advanced pancreas cancer survival.  This study is thorough with their patient selection/exclusion criteria, which adds to the utility of the study.   Moreover, while this study is largely correlative, the attempts to connect to underlying biology in the discussion are very useful. 

Minor comments to be addressed: 

1. Captions should go underneath of the table/figure to avoid confusion.  

2. In the discussion, the authors talk about hyperglycemia as a potential underlying mechanism.  Does FDG-PET correspond with worse outcomes in this type of patient population?  Moreover, section 2.4 "treatment response evaluation" says that patients underwent FDG-PET imaging; does GLUCAR correlate with FDG uptake? 

Author Response

Reviewer 2

We express our gratitude to Reviewer 2 for the invaluable comments, which will definitely enhance the comprehensiveness of our manuscript.

Comment 1. Captions should go underneath of the table/figure to avoid confusion.

Response 1. The required changes have been made in accordance with the journal's requirements.

Comment 2. In the discussion, the authors talk about hyperglycemia as a potential underlying mechanism. Does FDG-PET correspond with worse outcomes in this type of patient population? Moreover, section 2.4 "treatment response evaluation" says that patients underwent FDG-PET imaging; does GLUCAR correlate with FDG uptake?

Response 2. Several authors have studied how FDG-PET parameters affect the outcomes of patients with PAC who have received different therapies. For example, a recent study by Mokhtar and colleagues looked at the prognostic value of primary tumor volumetric parameters obtained from FDG PET/CT in the initial stage and their impact on OS and PFS for PAC patients (Mokhtar HM, Youssef A, Naguib TM, Magdy AA, Salama SA, Kabel AM, Sabry NM. The Significance of FDG PET/CT-Derived Parameters in Determining Prognosis of Cases with Pancreatic Adenocarcinoma: A Prospective Study. Medicina (Kaunas). 2022 Aug 1;58(8):1027. doi: 10.3390/medicina58081027.). Study results revealed that FDG uptake and the tumor glycolytic activity were substantially linked with a shorter PFS.          

Considering a potential correlation between the FDG uptake values and baseline GLUCAR levels, we analyzed this to address the reviewer’s query. However, we found no significant correlation between these parameters. Although this issue was beyond the scope of our investigation, we included this information as a further limitation of our study.

Reviewer 3 Report

Comments and Suggestions for Authors

The authors discussed a Novel Somay’s GLUCAR Index Efficiently Predicts Survival Outcomes in Pancreas Cancer Patients Receiving Chemoradiotherapy.

The article highlights the grim statistics of pancreatic adenocarcinoma, noting the very low 5-year survival rate. This underscores the critical need for advancements in treatment and early detection.

The study's findings that higher GLUCAR levels are independently associated with poorer progression-free survival and overall survival outcomes in LA-PAC patients undergoing definitive CCRT highlight the potential of GLUCAR as a significant prognostic marker.

Overall the manuscript is well written

the abstract needs some spaces between variables

The introduction could be clearer , by deleteing some details of the studies 

Results are well described

In the discussion I would add some comments to the authors : 

What mechanisms might explain the association between high GLUCAR levels and poorer outcomes in LA-PAC patients, is there any comparison with other studies in the literature ?

How can GLUCAR levels be integrated into current clinical workflows for the treatment planning of LA-PAC patients?

Could the authors compare their score with other existing prognostic markers in terms of predicting outcomes ?

Are there any known confounding factors that could affect GLUCAR levels?

Comments on the Quality of English Language

minor editing

Author Response

Reviewer 3

We express our gratitude to Reviewer 2 for the invaluable comments, which will enhance our manuscript's comprehensiveness.

Comment 1. The abstract needs some spaces between variables

Response 1. Space defects caused by editing problems are now carefully checked and corrected as recommended.

Comment 2. The introduction could be clearer by deleting some details of the studies 

Response 2. In the Introduction section, we discussed the details of only two studies, which directly emphasizes what we need to form a basis for our current study. We respectfully request to uphold their inclusion. Nevertheless, should it be deemed necessary, they can be omitted. 

Comment 3. In the discussion I would add some comments to the authors: What mechanisms might explain the association between high GLUCAR levels and poorer outcomes in LA-PAC patients, is there any comparison with other studies in the literature?

Response 3. Please refer to Response 5.

Comment 4. How can GLUCAR levels be integrated into current clinical workflows for the treatment planning of LA-PAC patients?

Response 4. ‘Our research findings emphasize the urgent need for implementing more advanced staging tools and more potent systemic therapies to the staging and treatment algorithms of these patients. In this context, liquid biopsies obtained from plasma or wash samples via endoscopic ultrasound-guided fine-needle biopsy may help detect high-risk LAPAC patients for early DM [64]. Besides this maneuver, it may be wise to initiate induction chemotherapy first and withhold aggressive CCRT for those who remain free of DM after systemic treatment. Upon further validation through research, GLUCAR may also ascertain the requisite intensity of therapies, potentially enhancing patient prognoses by tailoring treatments in some patients while averting futile interventions in others deemed resistant to presently available treatment modalities.’ This issue is now discussed in the revised manuscript.

Comment 5. Could the authors compare their scores with other existing prognostic markers in terms of predicting outcomes?

Response 5. The manuscript's third and fourth paragraphs discuss the outcomes of glucose, CRP, albumin, and CAR studies to support our current findings. Each component of the new GLUCAR seems to have distinct effects on the outcomes of PAC patients, whether evaluated individually or as a combined biomarker such as CAR.

Comment 6. Are there any known confounding factors that could affect GLUCAR levels?

Response 6. Numerous factors contribute to elevated CRP levels. These may encompass autoimmune conditions, such as rheumatoid arthritis and lupus, as well as specific types of inflammatory bowel disease, such as Crohn’s disease and ulcerative colitis. Pericarditis, infections, obesity, organ and tissue injury, cancer, and cancer cachexia are also implicated. Notably, an inverse relationship exists between CRP levels and albumin production in hepatocytes, leading to a reduction in albumin levels coincident with escalating CRP measures. Similarly, various factors can cause hyperglycemia. These factors include food and physical activity choices, dehydration, certain medications (especially those containing steroids), chronic pancreatitis, pancreatic cancer, infection, injury, surgery, stress, and hormonal changes. Conversely, several factors, including menstruation, celiac disease, alcohol intake, exercise, outside temperature, time of day, and timing of nutrition, may lead to a hypoglycemic state. Therefore, most of these factors were considered exclusion criteria in our study.